# Predictors of Tumour Growth and Autonomous Cortisol Secretion Development during Follow-Up in Non-Functioning Adrenal Incidentalomas

**DOI:** 10.3390/jcm10235509

**Published:** 2021-11-25

**Authors:** Marta Araujo-Castro, Paola Parra Ramírez, Cristina Robles Lázaro, Rogelio García Centeno, Paola Gracia Gimeno, Mariana Tomé Fernández-Ladreda, Miguel Antonio Sampedro Núñez, Mónica Marazuela, Héctor F. Escobar-Morreale, Pablo Valderrabano

**Affiliations:** 1Department of Endocrinology & Nutrition, Hospital Universitario Ramón y Cajal, 28034 Madrid, Spain; hectorfrancisco.escobar@salud.madrid.org; 2Instituto de Investigación Biomédica Ramón y Cajal (IRYCIS), 28034 Madrid, Spain; 3Universidad de Alcalá, 28801 Madrid, Spain; 4Department of Endocrinology & Nutrition, Hospital La Paz, 28046 Madrid, Spain; paola.parra@salud.madrid.org; 5Department of Endocrinology & Nutrition, Complejo Asistencial Universitario de Salamanca, 37007 Salamanca, Spain; crisrlz@hotmail.com; 6Department of Endocrinology & Nutrition, Hospital Universitario Gregorio Marañón, 28007 Madrid, Spain; rogelio.garcia@salud.madrid.org; 7Department of Endocrinology & Nutrition, Hospital Royo Villanova, 50015 Zaragoza, Spain; paogracia@gmail.com; 8Department of Endocrinology & Nutrition, Hospital Universitario de Puerto Real, 11510 Cádiz, Spain; marianatomefl@yahoo.es; 9Department of Endocrinology & Nutrition, Hospital Universitario de la Princesa, 28006 Madrid, Spain; msampeter@gmail.com (M.A.S.N.); mmarazuela.hlpr@salud.madrid.org (M.M.); 10Centro de Investigación Biomédica en Red Diabetes y Enfermedades Metabólicas Asociadas, 28029 Madrid, Spain

**Keywords:** adrenal incidentalomas, autonomous cortisol secretion, non-functioning adrenal incidentalomas, dexamethasone suppression test

## Abstract

Purpose: To assess the risk of developing autonomous cortisol secretion (ACS) and tumour growth in non-functioning adrenal incidentalomas (NFAIs). Methods: Multicentre retrospective observational study of patients with NFAIs. ACS was defined as serum cortisol >1.8 µg/dL after 1 mg-dexamethasone suppression test (DST) without specific data on Cushing’s syndrome. Tumour growth was defined as an increase in maximum tumour diameter >20% from baseline; and of at least 5 mm. Results: Of 654 subjects with NFAIs included in the study, both tumour diameter and DST were re-evaluated during a follow-up longer than 12 months in 305 patients. After a median follow-up of 41.3 (IQR 24.7–63.1) months, 10.5% of NFAIs developed ACS. The risk for developing ACS was higher in patients with higher serum cortisol post-DST levels (HR 6.45 for each µg/dL, *p* = 0.001) at diagnosis. Significant tumour growth was observed in 5.2% of cases. The risk of tumour growth was higher in females (HR 10.7, *p* = 0.004). Conclusions: The frequency of re-evaluation with DST in NFAIs during the initial 5 years from diagnosis can probably be tailored to the serum cortisol post-DST level at presentation. Re-evaluation of NFAIs with imaging studies, on the other hand, seems unnecessary in most cases, particularly if the initial imaging demonstrates features specific to typical adenoma, given the low rate of significant tumour growth.

## 1. Introduction

The increasing use of imaging studies has led to increasing diagnosis of adrenal incidentalomas (AIs) in recent years. It is estimated that 5% of the general population have AIs but the incidence increases with age [1]. In patients with AIs there are two important aspects that need to be ruled out at presentation: (1) malignancy, which is generally done by imaging characteristics; and (2) functionality [2]. Although most AIs are benign and non-functioning (NFAIs), about 15–30% are associated with hormonal hypersecretion. Autonomous cortisol secretion (ACS) is the most common functional alteration in AIs and it has been associated with an increased cardiometabolic risk [3].

During follow-up, 5% to 28% of NFAIs are expected to develop ACS, depending on whether ACS diagnosis is established by serum cortisol post-DST level above 5.0 µg/dL or 1.8 µd/dL, respectively; and 3% are expected to grow over 10 mm in maximum diameter [4]. Despite the known risk for ACS development and its associated increased cardiometabolic risk [5,6], it remains controversial whether NFAIs require long-term follow-up; or how closely they should be monitored. The European Society of Endocrinology and the European network for the study of adrenal tumours (ESE/ENSAT) [2] and the Italian Association of Clinical Endocrinologists (EMA) [7] guidelines consider it unnecessary to repeat hormonal or imaging evaluations in NFAIs if radiological features are typical of adenoma at presentation. On the other hand, under the same circumstances, the National Institutes of Health (NIH) [8], the French Endocrinology Society (FES) [9], the Spanish Society of Endocrinology and Nutrition (SEEN) [10] and the American Association of Clinical Endocrinologists (AACE/AASE) [11] guidelines recommend repeating a dexamethasone suppression test annually for up to 5 years and imaging studies for at least one year in tumours less than 4 cm or two years if they are 4 cm or larger.

The hypothesis of our study was that clinical, biochemical, and/or radiological features of NFAIs that remain stable during follow-up might be different from those NFAIs that experience significant tumour growth and/or develop ACS. Thus, the objective of our study was to identify predictors of ACS progression and tumour growth during the follow-up of NFAIs.

## 2. Materials and Methods

This study was approved by the Hospital Universitario La Princesa’s and Hospital Ramón y Cajal’s Ethics Committees (Madrid, Spain) (approval date: 23 September 2019, acta CEIm 10/19, approval number: 3702) and received funds from the Society of Endocrinology, Nutrition and Diabetes of Madrid (SENDIMAD) and from the Instituto de Investigación Biomédica Ramón y Cajal (IRYCIS).

### 2.1. Patients

In this retrospective study, we included patients between 18 and 90 years old with at least one AI equal to or greater than 1 cm evaluated between 2013 and 2020 at the Endocrine Department of one of the seven participating Spanish institutions. We excluded patients with: (i) known diagnosis of hereditary syndromes associated with adrenal tumours; (ii) chronic treatment with glucocorticoids or drugs that impair dexamethasone metabolism; (iii) current treatment with oral hormonal contraceptives (treatment should be suspended for at least 6 weeks before the functionality studies); and (iv) AIs identified during the extension study of an extra-adrenal primary cancer. For the present study, we excluded patients with missing values in the overnight 1 mg-dexamethasone suppression test (DST) or unavailable imaging studies (CT or MRI) at diagnosis or during follow-up; follow-up lower than 12 months; evidence of malignancy; or functioning adenomas at presentation (Figure 1). Study data were collected and managed using REDCap electronic data capture tools hosted at IRYCIS in Madrid, Spain [12,13]. REDCap (Research Electronic Data Capture) is a secure, web-based software platform designed to support data capture for research studies, providing (1) an intuitive interface for validated data capture; (2) audit trails for tracking data manipulation and export procedures; (3) automated export procedures for seamless data downloads to common statistical packages; and (4) procedures for data integration and interoperability with external sources.

### 2.2. Clinical and Hormonal Assessment

Medical records were reviewed to extract demographic information, comorbidities, body mass index (BMI) and systolic and diastolic blood pressure. Comorbidities possibly related to ACS were defined based on current standards, as we have previously published [14,15,16]. Cardiovascular disease was defined as ischemic heart disease or heart failure, and cerebrovascular disease as transient ischemic attack or acute stroke. At study entry, all AI patients were evaluated with at least DST and urinary free metanephrines or catecholamines. Other hormones, such as 8 am serum cortisol, adrenocorticotropic hormone (ACTH), dehydroepiandrosterone sulphate (DHEA-S), late-night salivary cortisol and 24-urinary free cortisol (UFC), were evaluated in some patients when the attending physician considered it necessary. Plasma aldosterone/renin activity or concentration ratio was also evaluated in hypertensive or hypokalaemic patients; and 17-hidroxyprogesterone and serum basal cortisol concentrations in patients presenting with bilateral AIs. Moreover, patients underwent routine biochemical profiles including fasting plasma glucose, total, LDL and HDL cholesterol levels, and triglyceride concentrations. HbA1c was measured in some patients at the discretion of the treating physician. We analysed patients’ data obtained during their initial evaluation and at their last available follow-up visit.

Serum cortisol was measured by competitive chemiluminescence in solid phase and electrochemiluminescence and immunochemiluminescence assays, intra-assay coefficient of variation (CV) was <10%; ACTH by electrochemiluminescence and sandwich type sequential immunoassay in solid phase, intra-assay CV was <10% and UFC by chemiluminescence assay of microparticles, chemiluminescence assay in immulite and in centaur with extraction in dichloromethane; intra-assay CV was <15%.

NFAIs was defined as an AI with serum cortisol post-DST below than or equal to 1.8 µg/dL with no evidence of other adrenal hormone excess. ACS development was diagnosed when serum cortisol post-DST was above 1.8 µg/dL at the last visit in the absence of specific clinical features of overt Cushing’s syndrome [3].

### 2.3. Radiological Investigation

In 185 patients, an initial evaluation was performed with abdominal computed tomography (CT), in 60 with magnetic resonance imaging (MRI) and in 60 with both techniques. The maximum adenoma diameter was used as tumour size. For bilateral AIs, the size of the largest adenoma was included in the analyses [14]. CT was repeated during follow-up in 164 patients, MRI in 115 and both studies in 26. Two hundred and forty four patients were re-imaged with the same imaging technique used at presentation, and with a different one in 61. Significant tumour growth was defined as an increase greater than 20% from the baseline measurement and of at least 5 mm [10,17]. Moreover, we reported the proportion of patients whose tumours grew more than 10 mm [4].

### 2.4. Statistical Analysis

Categorical variables are expressed as percentages; quantitative variables are expressed as mean ± standard deviation or median and interquartile range (IQR) depending on the normal distribution of the variable. Odds ratios (with 95% confidence intervals) and mean differences were calculated as association measures. For variables following the normal distribution, we used the Student’s t test to compare differences between two groups. Pearson’s correlation coefficients (r) were used to evaluate correlations between continuous variables. The chi-square test was performed for the comparison of categorical variables between independent groups. Predictive factors of ACS progression were identified by COX regression model. Nonparametric ROC curve analysis was used to determine the diagnostic accuracy of serum cortisol post-DST concentrations for the prediction of ACS progression in NFAIs. All statistical analyses were performed with STATA.15 (StataCorp. 2017. Stata Statistical Software: Release 15. College Station, TX, USA: StataCorp LLC). In all cases, a two-tailed *p* value < 0.05 or a hazard ratio (HR) with a 95% confidence interval, not including the null hypothesis, were considered statistically significant.

## 3. Results

### 3.1. Baseline Characteristics

A total of 305 patients with NFAIs met the inclusion criteria. At presentation, 24.4% (43/176) of NFAIs had ACTH below 10 pg/mL. 24 h-UFC was within reference ranges in all patients. Baseline characteristics are described in Table 1.

### 3.2. ACS Development during Follow-Up

After a median follow-up of 41.3 (IQR 24.7–63.1) months, 32 patients (10.5%) developed ACS. No patient developed overt Cushing’s syndrome. The incidence rate of ACS in NFAIs was of 19.3 cases/10,000 patient-year; and 60% of the cases occurred during the initial 5 years of follow-up. However, the other 40% of cases developed ACS afterwards. Patients with higher serum cortisol post-DST values at presentation had an increased risk of ACS development (Table 2 and Figure 2). Serum cortisol post-DST experienced a greater increase during follow-up compared to baseline values in patients with NFAIs that developed ACS than in those that continued to suppress cortisol below 1.8 µg/dL after the DST (0.8 ± 0.56 vs. 0.0 ± 0.34 µg/dL, *p* < 0.0001). Age and serum cortisol-post DST presented a positive, although weak, linear correlation (r = 0.16, *p* = 0.006). This correlation is clear when data are analysed, stratifying patients by age groups: <50, 50–59, 60–69, 70–79 and >80 years (Figure 3). Serum cortisol post-DST at diagnosis was the best predictor of ACS development during follow-up, almost doubling the risk at 0.45 µg/dL increments (Table 3). The area under the ROC curve (AUC) of serum cortisol post-DST at diagnosis for the prediction of ACS development was 0.69 (95% CI 0.63, 0.74) and the best threshold to predict ACS development was 1.4 µg/dL (sensitivity 59.4% and specificity 72.0%). Age was a poor predictor for ACS development during follow-up (AUC 0.55). No statistically significant differences were found in tumour growth during follow-up between NFAIs that developed ACS and those that did not (1.9 ± 7.6 vs. 0.1 ± 4.2 mm, *p* = 0.321).

### 3.3. Cardiometabolic Profile during Follow-Up

A total of 53 patients developed one or more new comorbidities during follow-up. The most common incident cardiometabolic disease was dyslipidaemia in 20.6% (*n* = 30) followed by hypertension in 11.0% (*n* = 16), obesity in 7.1% (*n* = 10) and type 2 diabetes mellitus in 5.7% (*n* = 12). Only four patients experienced new cardiovascular events and one patient an acute stroke. No clinical or hormonal predictors for the development of comorbidities were identified. We did not find differences in the risk of developing comorbidities between patients with NFAIs who remained hormonally stable and those progressing to ACS (*p* = 0.775) (Appendix A Table A1) or in their grade of control during follow-up (*p* > 0.05 for differences in plasma glucose, HbA1c, cholesterol, LDL-c, HDL-c, triglycerides, and blood pressure changes during follow-up between groups). However, we found that cortisol post-DST levels at diagnosis were significantly higher in patients with hypertension (1.24 ± 0.36 vs. 1.11 ± 0.36 µg/dL, *p* = 0.001), type 2 diabetes mellitus (1.28 ± 0.36 vs. 1.14 ± 0.37 µg/dL, *p* = 0.004), dyslipidaemia (1.24 ± 0.37 vs. 1.11 ± 0.36 µg/dL, *p* = 0.002) or a cerebrovascular event (1.54 ± 0.38 vs. 1.16 ± 0.36 µg/dL, *p* = 0.014).

### 3.4. Tumour Growth during Follow-Up

During the observational period, the mean tumour growth was 0.3 ± 4.70 mm. Significant tumour growth (>20% and at least 5 mm in maximum diameter) was observed in ten patients (5.2%). The median growth among these tumours was 14.0 ± 8.28 mm. There were seven patients with AIs that grew more than 10 mm during follow-up; one of which underwent adrenalectomy. This patient presented with bilateral AIs of 25 mm with low opposition in the phase signal in the MRI that grew 16 mm over 35 months, developed atypical radiological features during follow-up; and upon resection was diagnosed of metastases from colon cancer. There were no differences in follow-up time between patients with tumours that did and did not grow significantly (60.8 ± 52.6 vs. 49.0 ± 32.3 months, *p* = 0.494). Female sex was the only baseline feature associated with tumour growth (Table 4). The risk of significant tumour growth (>5 mm) was 10.5% in women and 1.0% in men.

Final tumour size was strongly correlated with initial (r = 0.78, *p* = 0.012) and diagnosis and last-visit serum cortisol post-DST levels (r = 0.15, *p* = 0.02 and r = 0.13, *p* = 0.039, respectively). ACS was developed during follow-up in 20.0% of tumours that demonstrated significant growth; and in 10.4% of tumours that remained stable in size, but differences were not statistically significant (*p* = 0.388).

## 4. Discussion

In this large multicentre retrospective observational study, 10.5% of NFAIs developed ACS and 5.2% grew significantly over a mean follow-up time of 41.3 months. Higher serum cortisol post-DST levels were found to be associated with ACS development; female sex was found to be associated with tumour growth; and serum cortisol post-DST levels were found to be linearly correlated with tumour size.

ACS, defined as serum cortisol post-DST greater than 1.8 µg/dL, was developed in 10.5% of NFAIs during follow-up. Results of previous studies are difficult to summarize, as several different diagnostic criteria for ACS development, follow-up time and cohort selection criteria have been used. Nonetheless, the rate of ACS in our study was within the expected range [18,19,20]. No patients developed overt Cushing’s syndrome in our series, which is also consistent with the very low rates reported in previous studies [4,21]. In this study, no statistically significant differences were observed in the risk of developing cardiometabolic comorbidities during follow-up between patients who developed ACS and patients who did not. However, this is probably a consequence of several factors: (1) a type 2 error, due to low rates of ACS and cardiometabolic comorbidities development; and (2) short follow-up period, as the development of comorbidities is likely directly associated not only with the serum cortisol post-DST level but also with the time of exposure. Several previous studies, however, have observed an increased cardiometabolic risk in patients with AIs and ACS [22,23]. Our data support this association as, despite the lack of statistically significant differences, a higher proportion of new comorbidities in patients who progressed to ACS than in patients who remained suppressible after DST was also observed (Table A1). Furthermore, the prevalence of hypertension, type 2 diabetes mellitus and dyslipidaemia in the entire cohort of this study was significantly lower than in the 337 patients excluded for meeting criteria for ACS at presentation (Table A2). Thus, because most cases developing ACS do so during the initial 5 years of follow-up (60% in this study), it seems reasonable to monitor NFAIs with 1mg-DST during this period. This might not be cost-effective, however, in the elderly with adequate serum cortisol suppression after 1 mg DST at presentation.

Identifying which patients are going to progress to ACS would allow individualizing follow-up. In this study, higher serum cortisol post-DST levels were associated with ACS development. ACS diagnosis is often challenging, and the best diagnostic threshold is still unclear. This is a consequence of autonomous cortisol secretion being a continuum that DST is unable to characterize in its milder forms. Thus, lower thresholds are more sensitive but are less specific than more stringent criteria. Nonetheless, an increased cardiovascular risk has been observed even for patients with NFAIs, that is, with serum cortisol post-DST equal to or below 1.8 µg/dL, in previous studies [24]. This might reflect the impact of mild glucocorticoid excess, which is currently unrecognized. Supporting this hypothesis, our study found that patients with cardiometabolic comorbidities had higher serum cortisol post-DST values than those without comorbidities. Moreover, patients with DST values closer to the ACS diagnostic threshold (1.8 µg/dL) were at increased risk for ACS development. However, these observations need further validation. Urinary steroid profiling could prove helpful in identifying these patients earlier in the future [25]. Until then, we think the serum cortisol post-DST levels at presentation could be used to individualize the follow-up of patients with NFAIs as described in Table 5. Measuring serum dexamethasone along with serum cortisol might help to identify false positive results of the test that are due to impaired dexamethasone absorption or accelerated metabolism of the drug. DHEAS, UFC and ACTH levels could also be repeated during follow-up when changes in DST values are detected. If serum cortisol post-DST levels continues to be stable and below 1.8 µg/dL after 5 years of follow-up, it seems reasonable to stop looking for ACS. However, periodic evaluation of other cardiometabolic factors might need to be continued given the increased risk in patients with NFAIs.

Older age was also found to be associated with higher serum cortisol post-DST levels in patients with AIs. It is possible that AIs with mild unrecognized hypercortisolism have had a longer time to progress in some older patients, and thus are closer to becoming apparent by impairing the DST results. This is also supported by the fact that, as follow-up time increases, the proportion of NFAIs developing ACS is higher. Other studies identified tumour size [17,18], bilaterality and low/suppressed ACTH values [17] to be predictive factors of ACS development in NFAIs. We did not find such an association but our results could suffer from a type 2 error given the low number of events.

During follow-up, 5.2% of tumours grew at least 5 mm and 2.3% grew more than 10 mm, which is within the expected range, as confirmed by a recent meta-analysis [4]. We did not find differences in the risk of growth according to the tumour size at presentation. A previous meta-analysis, however, found NFAIs of 25 mm or larger to have a lower risk of growth than smaller tumours, suggesting that AIs might grow until they reach a quiescent state [4]. To our knowledge, no other previous clinical study has identified female sex as a risk factor for tumour growth. However, as in other tumours, oestradiol enhances the progression and migration of endothelial cells in adrenal tumours [26]. In fact, the proliferation rate in steroidogenic cells of female rats is 6.3-fold higher than in male rats [27,28]. The correlation between serum cortisol post-DST levels and tumour size has been described previously in several studies [14,29,30]. We were unable to demonstrate an increased risk of ACS development in patients experiencing tumour growth during follow-up, but again the study could be suffering from a type 2 error, because serum cortisol post-DST levels were linearly correlated with the size of the AIs.

As reported in previous studies, the risk of malignancy in NFAIs with characteristics of adenoma on imaging studies is negligible despite growth [4]. These data support the ESE/ENSAT recommendations of not repeating radiological studies in AIs that measure less than 4 cm and look benign on imaging studies as the risk of malignant transformation is anecdotal. On the other hand, in AIs larger than 4 cm, a single radiological re-evaluation in 6–12 months might be enough to completely rule out malignancy [10].

Our study has some limitations, starting with the retrospective design, which limits the quality of the data; and does not allow the establishment of causality. The cohort likely suffers from selection bias, as all patients included in the study were evaluated at Endocrine departments of tertiary academic institutions. The follow-up period might be too short for some patients. We excluded patients with less than 1 year of follow-up, and the mean follow-up time of the study cohort was <5 years (41 months). Considering that we observed ACS development continuously during follow-up, it is likely that some AIs that will develop ACS in the future were analysed within the non-functioning group, potentially leading to an underestimation of patients developing ACS and diluting differences between groups. Thus, prospective studies with longer follow-up data are needed to confirm our observations. Some patients (*n* = 61) were re-imaged using a different imaging technique than the one used at presentation, which could lead to under or overestimation of the incidentaloma’s growth. However, to minimize this limitation, we only considered changes in tumour size greater than 20% to be significant. Moreover, we have not evaluated the cost effectiveness of our proposed approach to follow-up, which needs prospective validation.

## 5. Conclusions

The re-evaluation of NFAIs with DST for at least 5 years seems appropriate given that most cases of ACS development occur during this period. However, the frequency of such evaluation can probably be tailored to the serum cortisol post-DST level at presentation. The re-evaluation of NFAIs with imaging studies, on the other hand, seems unnecessary, particularly if the initial imaging demonstrates features specific to typical adenoma, given the low rates of tumour growth.

## Figures and Tables

**Figure 1 jcm-10-05509-f001:**
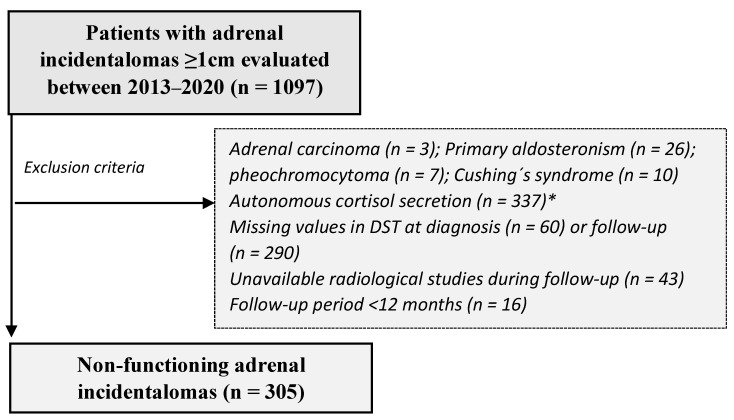
Study population. DST = dexamethasone suppression test; * Autonomous cortisol secretion was defined as post-DST serum cortisol above 1.8 µg/dL in the absence of specific clinical features of overt Cushing’s syndrome.

**Figure 2 jcm-10-05509-f002:**
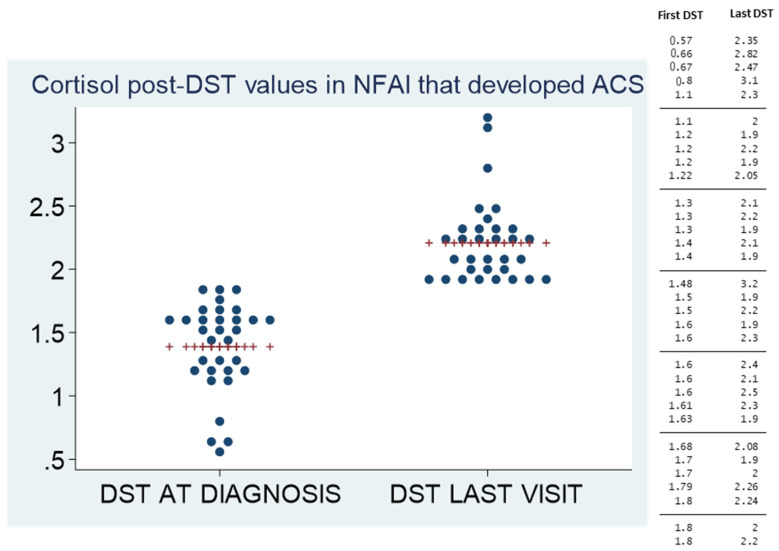
Evolution of the cortisol post-DST at diagnosis and at the last visit in patients who developed ACS. The values of cortisol post-DST at diagnosis and in the last visit in patients with NFAIs who developed ACS are described in the table and figure, including the mean value (red marker) in each moment.

**Figure 3 jcm-10-05509-f003:**
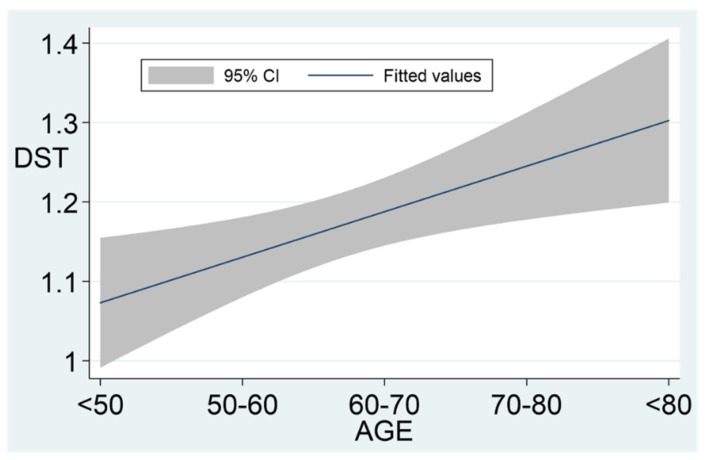
Association between age and DST values. DST: dexamethasone suppression test; DST levels increased by age: mean levels by groups were the following: <50 yo (1.0 ± 0.32 µg/dL), 50–60 yo (1.1 ± 0.38 µg/dL), 60–70 yo (1.2 ± 0.36 µg/dL), 70–80 yo (1.2 ± 0.37 µg/dL) and >80 yo (1.2 ± 0.43 µg/dL).

**Table 1 jcm-10-05509-t001:** Baseline characteristics of study population (*n* = 305).

Variable	Value
Age (years)	61.5 ± 10.2
Any ACS-related comorbidities	76.4%
Hypertension	47.2%
Type 2 diabetes mellitus	24.3%
Dyslipidaemia	46.9%
Obesity	39.3%
Cardiovascular disease	9.8%
Cerebrovascular disease	2.0%
Fasting plasma glucose (mg/dL)	105.9 ± 5.87
HbA1c (%) (*n* = 110)	6.6 ± 5.87
LDL-c (mg/dL) (*n* = 225)	119.2 ± 35.39
HDL-c (mg/dL) (*n* = 225)	52.4 ± 15.69
Triglycerides (mg/dL) (*n* = 278)	115.5 ± 55.34
1 mg DST (µg/dL)	1.2 ± 0.4
24 h-UFC (µg/24 h) (*n* = 142)	44.1 ± 81.25
ACTH (pg/mL) (*n* = 176)	19.1 ± 13.54
DHEAS (µg/dL) (*n* = 183)	530.6 ± 604.43
Bilateral tumours	20.1%
Tumour size (mm) (*n* = 380)	18.3 ± 7.33

DST = dexamethasone suppression test; DHEAS = dehydroepiandrosterone-sulphate (reference range was sex and age dependant); 24 h-UFC = 24 h urinary free cortisol; For ACTH and UFC different reference ranges were used depending on the local laboratory of the different hospitals.

**Table 2 jcm-10-05509-t002:** Risk factors of ACS development in NFAIs (*n* = 305).

Variable	HR (95% CI), *p* Value
Female sex	0.56 (0.24, 1.28); *p* = 0.155
Age at diagnosis (years)	1.04 * (1.00, 1.08); *p* = 0.065
ACS related comorbidities	1.42 (0.54, 3.75); *p* = 0.464
1mg DST (µg/dL)	6.44 * (1.88, 22.05); *p* = 0.001
UFC (µg/24 h)	1.00 * (0.98, 1.03); *p* = 0.609
ACTH (pg/mL)	0.99 * (0.94, 1.04); *p* = 0.664
DHEAS (µg/dL)	1.00 * (1.00, 1.00); *p* = 0.897
Tumour size (mm)	0.99 * (0.93, 1.04); *p* = 0.627
Bilaterality	1.35 (0.51, 3.56); *p* = 0.560

DST = dexamethasone suppression test; DHEAS = dehydroepiandrosterone-sulphate; UFC = urinary free cortisol. * Per each increased unit.

**Table 3 jcm-10-05509-t003:** Incidence of ACS development based on the serum cortisol post-DST level at diagnosis (follow-up time: 41.3 (IQR 24.7–63.1) months).

DST Group	Cases	Cumulative Incidence (95% CI)
≤0.45 µg/dL	0/5	0.00 (0.00–0.43)
0.45–0.9 µg/dL	4/83	0.05 (0.02–0.12)
0.9–1.35 µg/dL	9/117	0.08 (0.04–0.14)
1.35–1.8 µg/dL	19/100	0.19 (0.13–0.28)
Total	32/305	0.10 (0.08–0.14)

DST = dexamethasone suppression test. MH Test for linear Trend: Chi^2^ = 10.65 (*p* = 0.0011).

**Table 4 jcm-10-05509-t004:** Risk factors of tumour growth in NFAIs (*n* = 305).

Variable	HR (95% CI), *p* Value
Female sex	10.71 (1.34, 85.71); *p* = 0.004
Age at diagnosis (years)	1.05 * (0.97, 1.12); *p* = 0.201
1mg DST (µg/dL)	4.12 * (0.54, 31.51); *p* = 0.152
24 h-UFC (µg/24 h)	0.99 * (0.91, 1.07); *p* = 0.783
ACTH (pg/mL)	0.93 * (0.79, 1.11); *p* = 0374
DHEAS (µg/dL)	1.00 * (0.99, 1.00); *p* = 0.132
Tumour size (mm)	0.94 * (0.84, 1.05); *p* = 0.224
Tumour size < 25 mm	0.29 (0.08, 1.05); *p* = 0.076
Bilaterality	1.64 (0.34, 7.90); *p* = 0.558

DST = dexamethasone suppression test; DHEAS = dehydroepiandrosterone-sulphate; 24 h-UFC = 24 h urinary free cortisol; * Per each increased unit.

**Table 5 jcm-10-05509-t005:** Proposed follow-up for NFAIs based on serum cortisol post-DST levels at diagnosis.

DST Value at Diagnosis	Risk of ACS in Five Years	Suggested Follow-Up Recommendation
<0.9 µg/dL	5.8%	DST 5 years after diagnosis *
0.9–1.45 µg/dL	7.1%	DST every 2.5 years after diagnosis for 5 years *
1.45–1.8 µg/dL	19.8%	DST yearly after diagnosis for 5 years *

* Consider measuring ACTH, UFC and/or DHEAS depending on their values at diagnosis. Risk refers to the follow-up period of this study (41.3 (IQR 24.7–63.1) months).

## Data Availability

The data presented in this study are available on request from the corresponding author. The data are not publicly available due to privacy reasons.

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
