# Peer review of "Predictors of Tumour Growth and Autonomous Cortisol Secretion Development during Follow-Up in Non-Functioning Adrenal Incidentalomas"

_jcm, 2021, doi:10.3390/jcm10235509_

Round 1

Reviewer 1 Report

The authors of this paper retrospectively reviewed a consistent cohort of patients with adrenal incidentaloma with complete suppression after 1 mg dexamethasone test and re-assessed the test after a mean follow-up of 41 months. They diagnosed autonomous cortisol secretion in 10.5% of their cohort; the risk of develop ACS was directly correlated with the result of baseline test, confirming that this condition is a continuum that cannot be identified by an established cut-off. They also repeated adrenal imaging during follow-up finding an increase in tumor size in a low proportion of case. The topic faced is interesting, as no clear indication on how to follow these patients comes from the scientific societies. Despite this, I have some concerns:

  • First, authors considered a mean follow-up period of 41 months, but a significant proportion of patients were evaluated before this time-lapse. This fact might result in an underestimation of patients potentially developing ACS either within or after 5 years of follow-up. A minimum length of follow-up should be established in order to have more solid results. Please comment on that.
  • The proportion of patients developing ACS is 60% at line 60 and becomes 70% at line 244. Please reconcile data.
  • The age-range of patients included is very wide; is it really useful to repeat hormonal assessment in patients aged > 80 years with a normal cortisol suppression at baseline?
  • The development of cardiometabolic complications during follow-up should consider the effect of time, which is quite relevant considering the mean age at diagnosis and the fact that most patients had several features of metabolic syndrome at baseline
  • Similarly, as no differences in the comorbidities were observed between ACS and NFAI during follow-up, what is the sense of repeating 1mg-DST?
  • I was wondering whether patients initially evaluated with CT or MRI were followed using the same radiologic exam. If not this point should be mentioned as a limitation because it might resulted in under/overestimation of incidentaloma’s growth.
  • As dexamethasone absorbtion might hamper the result of the test, I would suggest the authors to discuss the possibility of dexamethasone measurement as a potential tool to confirm the results obtained especially considering that ACTH levels were not available in all patients.

Minor points:

-table 1: please indicate normal range for UFC, DHEAS and ACTH

-there is a typo at page 5, line 169: “ant”

Author Response

Dear reviewer,

Thank you very much for your sound and constructive comments and for giving us the opportunity to review and improve our manuscript

You can find a detailed point by point answer to your comments and concerns in the attached file

Reviewer 2 Report

Please see the attached review.

Author Response

Dear reviewer,

Thank you for your comments and for give us the opportunity to review our article.

You can find the detailed answers to your comments in the attached file

Round 2

Reviewer 2 Report

In my opinion the Authors significantly improved the manuscript.